# Acoustic Emission Characteristics of Graded Loading Intact and Holey Rock Samples during the Damage and Failure Process

**Xiaofei Liu** [1,2,*] , **Huajie Zhang** [1,2,*], **Xiaoran Wang** [1,2], **Chong Zhang** [1,2], **Hui Xie** [1,2], **Shuai Yang** [1,2] and **Weikai Lu** [1,2]

[1] Key Laboratory of Gas and Fire Control for Coal Mines (China University of Mining and Technology), Ministry of Education, Xuzhou 21116, China; xrwang1992@cumt.edu.cn (X.W.); zhangchong@cumt.edu.cn (C.Z.); xiehui@cumt.edu.cn (H.X.); shuaiyang@cumt.edu.cn (S.Y.); lwk920@163.com (W.L.)

[2] School of Safety Engineering, China University of Mining and Technology, Xuzhou 221116, China

[*] Correspondence: liuxiaofei@cumt.edu.cn (X.L.); zhanghjsafety@163.com (H.Z.)

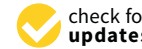

**Featured Application: This article describes various applications of in acoustic emission (AE) monitoring in the context of rock health monitoring in mines and engineering.**

**Abstract:** Rock burst is the result of the development and extension of micro-cracks during the loading process of large-scale rock mass in underground space engineering. Dynamic monitoring results by acoustic emission (AE) can accurately perceive the inner fracture evolution of rock mass and effectively warn about its induced disasters early. By contrastive testing the AE parameters in the whole fracture process of the intact and holey rock samples under graded loading, their spatiotemporal evolution rules were analyzed in this paper, and the damage model of rock samples based on AE localization events was established to analyze the relationship between rock damage and loads. The results show that: (1) Under the condition of grading loading, AE parameter increases with the increase of axial stress and show three states, respectively, which are slow-growth, stabilization and rapid increasing; meanwhile, the damage of the sample has a cumulative effect with time. (2) The AE counts and energy are highly correlated with the fracture of the sample that the more severe the damage of the sample, the faster the crack propagation as well as the higher the acoustic emission counts and the energy amplitude. The damage state of granite sample can be accurately judged by two parameters to character the damage evolution process and fracture mechanism. (3) Compared with the intact rock sample, due to the pressure relief effect of the hole, the rock sample containing the hole takes a long time in the compaction stage and with higher load stress level. Although the AE counts and energy were lower in the damage process, the general law of their response during damage and instability process still exists.

**Keywords:** graded loading; acoustic emission; precursory characteristics; fracture; holey granite sample

---

## 1. Introduction

Underground space engineering has entered a high-speed construction period with the development of deep excavation technology, but more and more underground engineering (such as mining/tunnel engineering) is prone to structural instability and result in disasters (such as rock burst/coal bump) due to the complex crustal stress conditions and lithological structures in the construction process [1,2]. The development and the expansion process of the surrounding rock internal fissures under the complicated loading conditions of the underground space are closely related

to its structural instability disasters. The stability monitoring has become the focus of engineering construction at home and abroad [3–5].

The gestation and occurrence of rock instability disasters in deep rock engineering is fundamentally the evolution process of internal primary crack closure, new crack generation and expansion of the large-scale rock under loading. As a concomitant phenomenon of releasing strain energy to the outside world in the form of elastic wave during the process of deformation or fracture of loaded rock material, the acoustic emission (AE) signal is able to monitor the failure process of sample timely, dynamically and continuously and to conduct three-dimensional localization of fracture event. Therefore, it is an effective tool for studying the material internal damage degree, fracture evolution and macroscopic cracks generation and expansion [6–8]. Stefano et al. [9] used AE signals to analyze the development of the fracture during the force rise phase and the fatigue damage characteristics of the static load phase, and monitored the development of the crack before the damage becomes visible. Liu et al. [10] used AE technology to timely record the process of crack generation, expansion, penetration and failure under uniaxial compression. The results were found that the AE energy and localization can timely reflect the whole expansion, penetration and failure process of the crack samples, which leads to better understanding the damage mechanism of rock mass in underground engineering. The fracture process of rock roadway by AE ringing count, energy, main frequency and other parameters were analyzed by Zhang et al. [11], which showed that the AE signal was closely related to the expansion stability and the crack size of rock crack. Mao et al. [12] monitored the AE signals of different kinds of microscopic particles during the cracking process of the samples and found that AE signal was closely related to the micro-structure of the particles during the crack propagation and expansion process. These above studies have shown that the occurrence of rock dynamic disasters is the result of the evolution of the internal fissures of the rocks, and has nonlinear characteristics in time; the changes of the parameters of the AE signals are not only related to the mechanical state of the rocks, but also to loading form, deformation field evolution, local crack expansion, macroscopic crack generation, etc. [13–16]. How to better invert and characterize the whole process of propagation, failure and instability of the crack of the loaded rock by the characteristics of the AE parameters and the variation, and warn about the occurrence of dynamic disasters in advance during the critical instability stage, were always the research direction of the rock AE application research.

Based on these, in this paper, intact and holey rock samples were simulated different types of underground space structure, and the graded loading was simulated the increasing stress conditions of the rock mass induced by the overburden rock layer and underground mining. The characteristics of AE signals were tested during the process of load-damage of two kinds of rock samples, and their differences were also found. Meanwhile, combining with the results of the AE localization, the damage evolution process of the sample and the special behavior of AE signals in the sub-stability phase were also analyzed. This research results can provide an important experimental basis for the rock damage evaluation and the precursor identification of rock mass instability.

## 2. Experimental Systems, Samples and Programs

### 2.1. Sample Preparation

The samples were all granite and taken from the Huaibei mine in Anhui Province. Large-scale granite rock was collected from the construction site and was sent to the processing plant for initial processing to make a $100 \times 100 \times 100$ mm$^3$ square rock sample by hydraulic cutting machine. Some of these intact rock samples were selected and a cylinder of 20 mm diameter was taken out from the selected sample center by the hole punching machine to make a $100 \times 100 \times 100$ mm$^3$ square sample with a 20 mm diameter hole. The sample surfaces were polished and smoothed by a grinder and could be in good contact with the end face of the AE transducer during the AE test. Finally, rock samples were dried and then labeled and stored. In order to minimize the influence of the sample's heterogeneous dispersion on the experimental results, nine intact rock samples and 12 cavernous rock samples with

no crack on the surface and similar p-wave velocity (Figure 1) were selected by ultrasonic velocity measurement in advance.

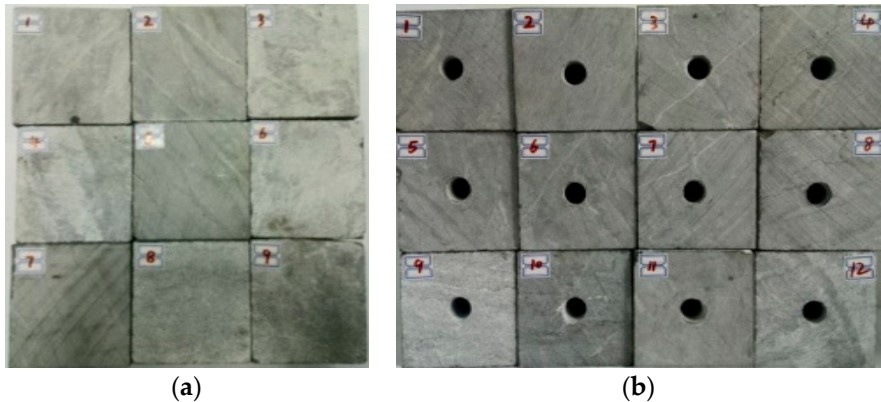

| (a) | (b) |
|-----|-----|

**Figure 1.** Experimental samples. (**a**) intact rock samples; (**b**) holey rock samples.

## 2.2. Experimental System

The experimental system mainly includes a loading system and an AE acquisition system (Figure 2). The loading system is a new YAW-4306 type electro-hydraulic servo pressure testing machine (mistras, MTS - SANS Shenzhen Operations, Shenzhen, China) by which mainly composed of an oil pump and a control system. The control system consists of DCS controller and Power Test software (Version 3.3; MTS—SANS Shenzhen Operations, Shenzhen, China), and can realize multiple functions such as force control, displacement control and force retention with up to 300,000 yards' resolution and ±1% relative error. The maximum load of the compressor oil pump can reach 3000 kN. Experimental system can do experiments such as stretching loading, uniaxial loading, grading loading, cyclic loading and unloading.

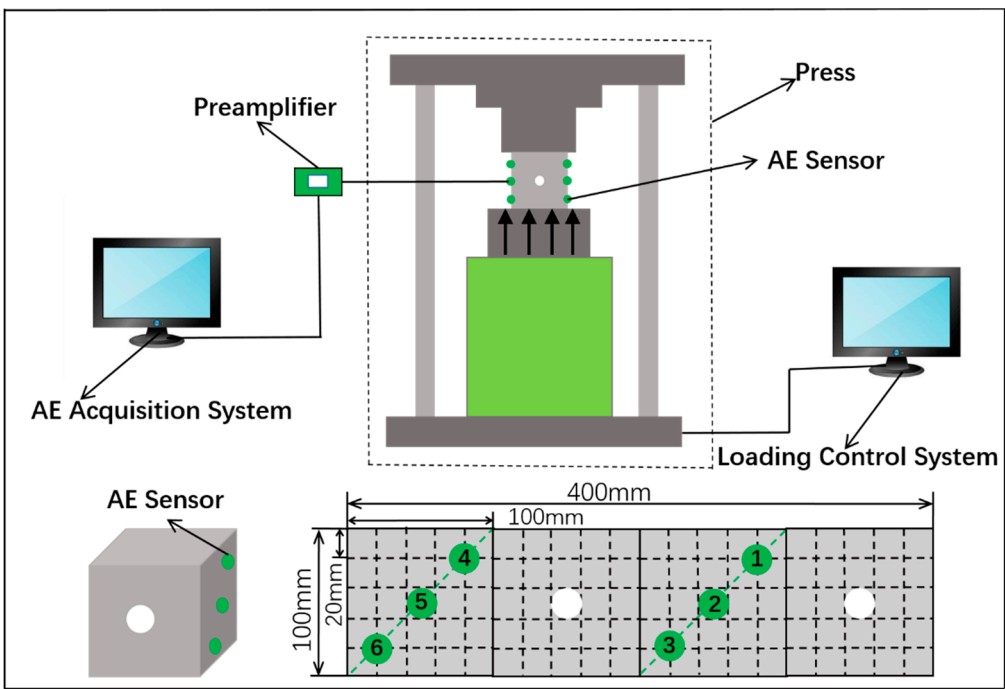

**Figure 2.** Schematic diagram of the experimental system.

The AE system adopts the Micro-II type AE monitoring host of Physical Acoustics Corporation. (195 Clarksville Rd, Princeton Junction, NJ, USA), which can engage data acquisition by up to

24 channels. Real-time acquisition of AE time domain parameters, original waveform data and spectrum analysis and spatially locate three-dimensional positioning of AE events can be realized through the matched Express-8 type rock test for data acquisition analysis software.

This experiment was carried out in the electromagnetic shielding room which can effectively isolate external electromagnetic interference and environmental noise, and the shielding effect can be up to 85 dB. In the experiment process, the movement of personnel was strictly controlled to ensure the experiment can be carried out in the condition of no interference to the greatest extent. Considering the electromagnetic noise interference, the sensor connectors were fixed on the test bench to avoid the flexing or twisting with coaxial-cable in the process of the experiment and the grounding wires were connected with experimental instrument to reduce unnecessary interference caused by electromagnetic noise during the acquisition of AE signals.

### 2.3. Experimental Calibration and Sensor Placement

According to the experimental scheme, the six resonant AE sensors (model R15$\alpha$, peak frequency 150 kHz) are fixed at the planned position on the surface of the sample through a bi-component glue before the experiment. The AE probes evenly were arranged along the diagonal of the two corresponding faces on the surface of the sample (holey sample should avoid AE probes arrangement on the surface with hole), which as shown in Figure 2.

In order to ensure the positioning accuracy, the lead-breaking experiment was used to test the accuracy and coupling of the positioning system before the experiment. The definition value and event locking value of AE events were adjusted repeatedly until the error distance between the location of the test point and the registration point was less than 2 mm as well as the response amplitude of each sensor was more than 90 dB. During the experiment, the preamplifier amplification gain was set as 40 dB, the threshold value as 100 μv, and the sampling rate as 1 MSPS.

### 2.4. Experimental Steps

(1) Selecting the rock samples (holey sample and intact sample) on the press platform and coating the coupling agent on the surface of the AE sensor in contact with the rock sample, the AE sensor was arranged on the opposite surfaces of the sample for signal acquisition.

(2) Connecting the data acquisition device and the AE probe before checking the correctness of the connection line and the instrument, and then set the relevant parameters.

(3) After starting the press and data acquisition device, the pressure platform was lift to make the rock sample contact with the upper surface of the indenter, and went through the grading loading test. The loading rate was initially 300 N/s, and every 3.0 MPa was set as a stage with 150 s force keeping time of each stage is. The AE test was suspended for the ultrasonic test in the middle process about 50 s, where the collected data and the press will stop acquisition until the sample was broken.

(4) When the experimental data were stored, they organized the instrument to carry out the next test.

## 3. Experimental Results

### 3.1. AE Timing Characteristics of the Grading Loading Process

During the micro-fracture process of granite loading, it will be accompanied by AE signals. Various changes will occur due to the unlike internal cracks under different stress levels of grading loading and each fracture corresponds to different levels of AE signals. From the test results of several rock samples, due to the small size of the sample, the AE signals received by the AE sensors at different positions show similar regularity in the timing characteristics, so the intact rock sample was analyzed in accordance with the 5-channel AE sensor (the hole rock sample uses the same channel of AE information). Figure 3 is a time-series response characteristic of the 5-channel AE sensor receiving AE signals of the two types of rock samples in the compact and elastic and failure stages under the staged

loading. The AE count in the figure mainly reflects the frequency of micro-fracture of the sample under load deformation, and the AE energy represents the energy released during the fracture of the sample.

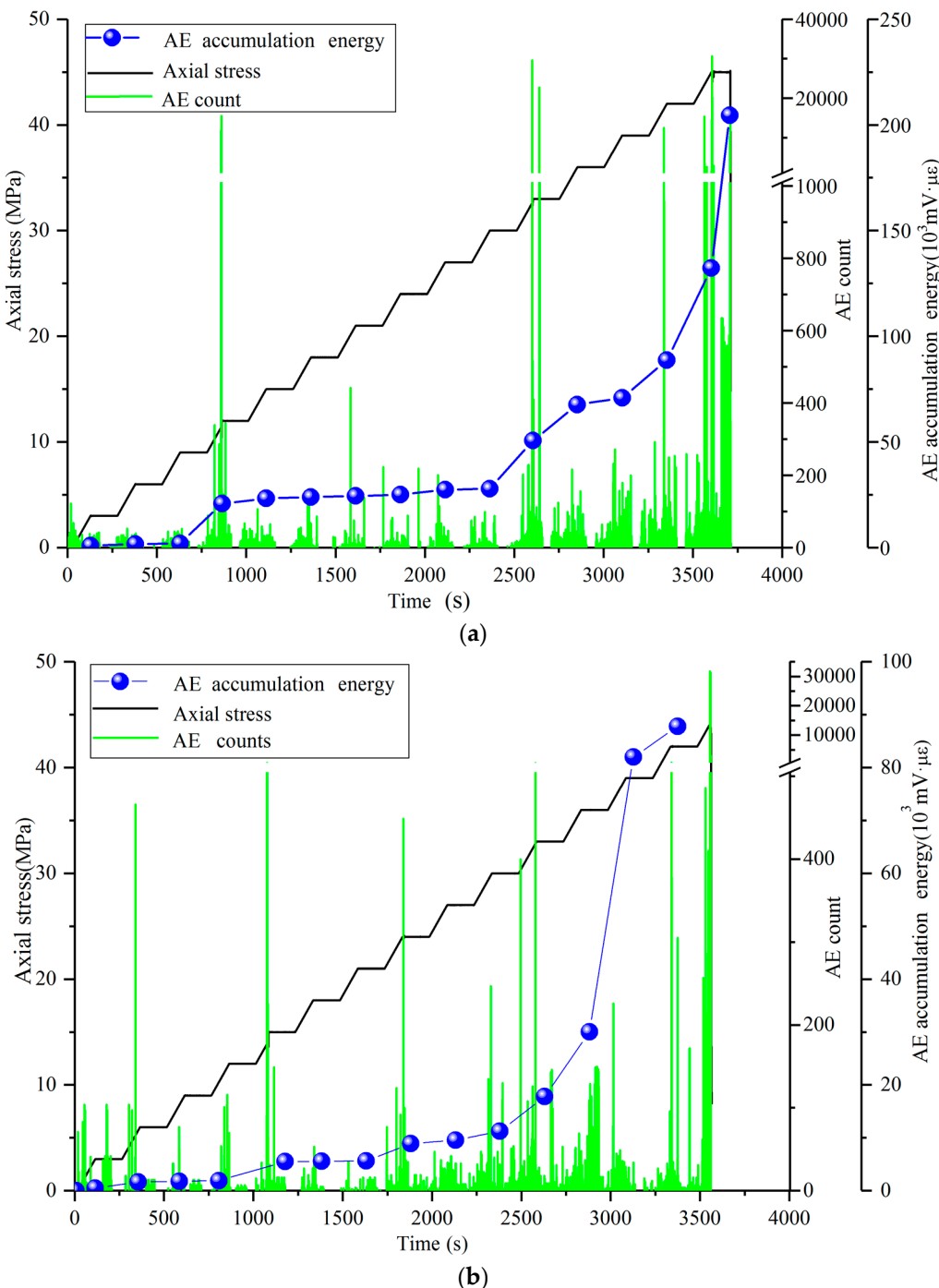

**Figure 3.** Sequential characteristics of granite AE parameter under grading loading. (**a**) intact rock sample; (**b**) holey rock sample.

It can be seen from Figure 3a that in the initial stage (0–9 MPa) of the staged loading of the intact sample, the AE count and the energy value are small due to the higher strength of the granite sample and the lower internal crack closure. As the grading loading progresses, the internal compactness of the sample increases, and the original fissures become fewer and fewer. At the same time, the stress level is insufficient to generate new fissures, so the AE count and energy show a slight downward trend. As the stress continues to increase, when the stress is in the rising phase of 9–12 MPa, the AE

signal has a high value response and then the sample enters into the elastic phase (12–32 MPa). At this time, micro-cracks begin to be generated inside the sample, and the generation of new fissures is relatively random but the number is relatively stability, while the cumulative energy of AE is also in a relatively stable phase. When the stress is in the rising phase of 30–33 MPa, the AE signal jumps again, for the sample from the elastic phase into the plastic phase (33–45 MPa), and begins to be irreversible deformation. Meanwhile, the micro-crack gradually expands and forms a small regional fracture and nucleation, which leads the AE count and energy all show a significant upward trend and significantly exceeded the first two stages. The cracks also expand and close with the increase of time in the force retention phase, the AE count and the cumulative energy rise faster, and the rheological characteristics of the sample become more and more obvious due to loading damage factors. When the sample reaches the damage fracture limit stage (42–45 MPa), the internal crack of the sample penetrates a lot and forms the main macro-crack, the AE count increases rapidly to the maximum value. When the stress level reaches retention phase (46 MPa), the sample completely cracks and destabilizes, the AE counts and energy increase rapidly and reach a maximum.

Compared with the intact granite sample, the AE counts and the energy value of the holey granite sample are lower under the same axial loading, but the general rule still exists. At the initial stage of loading (0–6 MPa), due to the action of the hole, the original crack of the sample is closed less, the regional response of the AE is high, and the energy level is low. The hole in the center of the sample generates to some extent the pressure relief by the axial load and it makes the compaction phase of the sample take a long time. When the stress is in the rising stage of 12–15 MPa, the AE count suddenly increases, the cumulative energy is sensitive, the new crack begins inside the sample, and the sample enters the elastic stage. In the early stage of the sample, due to the action of the hole there are many strains in the compaction stage. When the stress increases to 15–24 MPa, new cracks begin to form inside the sample, the amount is relatively low and stable, and the cumulative energy value of AE is relatively stable with graded loading. Compared with the intact sample, the holey sample advances into the shaping stage (27–42 MPa). Because the sample is subjected to irreversible damage which causes a large number of new cracks to expand and penetrate and destroys the sample locked bodies, the macroscopic fracture become obvious, and the sample is quickly destabilized.

*3.2. AE Response Characteristics of Damaged Samples during Loading*

According to the AE energy time-series characteristics and the three-dimensional positioning results of AE events in the rock sample loading process (as shown in Figure 4), the AE energy response characteristics of the loaded rock sample in the process of crack propagation were analyzed. It can be seen that, as the stress level increases, there is no linear relationship between the number of AE events inside the sample and the stress level of the sample. The key feature (the transition points that the sample loaded in three stages) of the internal crack propagation of the sample with different structures shows similarity at the stress level. It can be seen from the loading process of the intact rock sample that, in the previous compaction phase (0–866 s), there are many AE events and the local agglomeration phenomenon is obvious. At this time, the AE energy has a low value response. Then, the sample enters the elastic stage (866–2606 s) and the inside of the sample begins to produce micro-cracks. New fractures are mainly generated in the compacted region of the first stage, and the number of AE events is less than the number of AE events in the compaction stage. At this time, the overall increase of AE energy amplitude is not obvious with random energy jumps slightly. At the late loading stage (2606–3704 s), the internal cracks of the sample expanded and penetrated to form regional instability. According to the location of AE events, the AE events in this stage transferred from the geometric center of the sample to the outside. The overall amplitude of AE energy increased significantly and showed a significant jump when the sample was critically unstable.

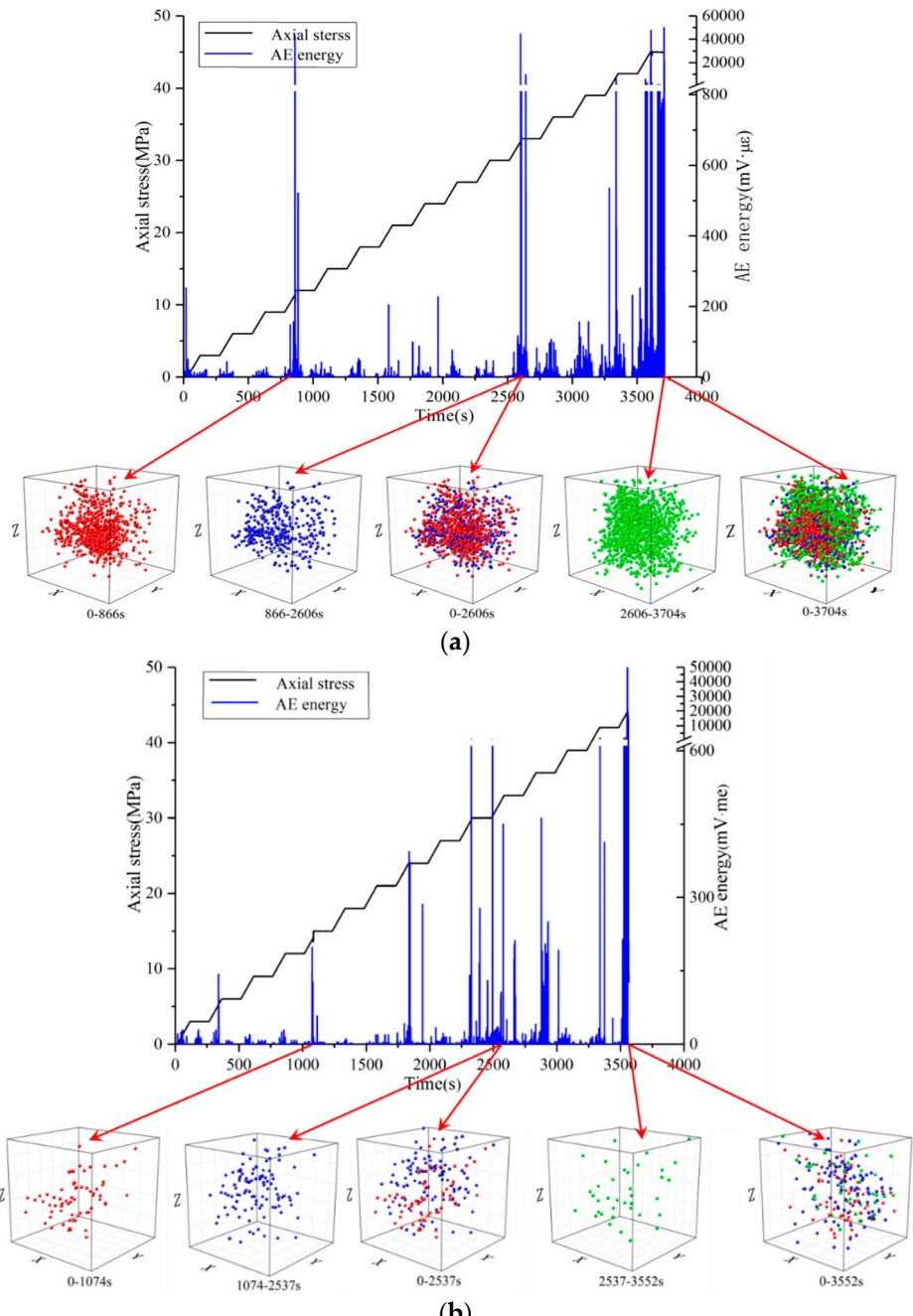

**Figure 4.** Time-space response of AE energy during damage of granite samples. (**a**) intact sample; (**b**) holey sample.

Compared with the intact sample, the number of AE localization events of the holey sample during loading is small, but the general law of energy fluctuations still exists when the sample is in the compaction stage (0–1000 s), the elastic stage (1000–2537 s), and the fracture instability stage (2537–3552 s). At the initial stage of loading (compaction stage), the AE localization event is randomly distributed around the geometric center of the sample and the hole effect is obvious, at this time, the AE energy generated by the sample damage is low overall and the phenomenon that energy suddenly jumps is not obvious. As the loading progresses, the AE energy of the sample is at a stable low value in the early stage of the elastic phase, and when the load level reaches 24 MPa, the AE energy suddenly jumps and drops and has a huge fluctuation partly. The less AE localization event corresponds to a

large AE energy value and the AE energy reaches the maximum value before the instability in the fracture instability stage (27–42 MPa) of the rock sample.

## 4. Discussion

### 4.1. Granite Instability Locked Bodies Fracture Nucleation Mechanism

As a kind of original heterogeneous material, the rock contains many structures with different strengths and scales due to the formation conditions, different mineral composition, burial conditions, geological tectonic movement and other factors, among them the structure with high strength controlling the stability of the sample which called locked bodies [17]. The process of macroscopic cracking and expansion of granite samples are the process of continuous fracture of locked bodies inside the rock mass. Combined with the AE localization map (Figure 4), it can be seen that the fracture process of the first locked solid of granite also undergoes a series of microscopic plastic yielding processes such as micro-fracture generation, expansion, and nucleation. At the microscopic scale, the micro-fracture rate inside the rock mass under load can be obtained by the Arrhenius Equation:

$$v_c(t) = A \exp(\frac{A_1 E \int_0^t \varepsilon(t)dt - U_0}{kT}),$$

(1)

where in: $v_c$ is the micro-fracture rate; $A$, $A_1$ are constant; $T$ is the absolute temperature of the rock mass; $k$ is the Boltzmann constant; $U_0$ is the fracture activation energy; $\dot{\varepsilon}(t)$ is the strain rate; and $E$ is the elastic modulus.

In order to reduce the influence of unnecessary factors, it can be considered that process is isothermal ($T$ is a constant), when the rock mass is subjected to load, and the strain loading rate is constant ($\dot{\varepsilon}(t) = \dot{\varepsilon}$). Then, when the sample is loaded to time $t$, the cumulative micro-fracture count can be shown as:

$$V_c(t) = \int_0^t v_c dt = \frac{kTA}{A_1 E \dot{\varepsilon}} \exp(-\frac{U_0}{kT})[\exp(\frac{A_1 E \dot{\varepsilon} t}{kT}) - 1].$$

(2)

As the loading progresses, $V_c$ increases exponentially. According to the fracture density criterion proposed by Zhurkov et al. [18], it is considered that, when the number of micro-fractures in the area around the locked bodies reaches the fracture limit, the locked bodies will break:

$$V_c(t)\big|_{r \leq A_2 r_i} \geq \frac{N_{si}}{\frac{4}{3}\pi r_i^3},$$

(3)

where in: $r_i$ is the radius of the *i*th locked solid; $V_c(t)\big|_{r \leq A_2 r_i}$ is the same sphere as the *i*th locked bodies, the radius is the total number of micro-fractures in the $A_2 r_i$ sphere, wherein $A_2$ is a constant, $A_2 > 1$; $N_{si}$ is the critical point of the *i*th locked bodies fracture threshold.

Joint Equation (3), the critical stress of the *i*th locked bodies fracture is:

$$\sigma_{ci} = \frac{kT}{A_1} \ln\left[\frac{3N_{si}A_1 E \dot{\varepsilon}}{4\pi r_i^3 kTA} \exp(\frac{U_0}{kT}) + 1\right].$$

(4)

Since the sample is a regular cube, the geometric center of the sample is most prone to stress concentration and the crack in this area first develops. When the surrounding defect density exceeds the critical threshold of the locked bodies fracture (as known from Equation (3)), the locked bodies in geometric center of granite sample firstly yield to fracture, and the AE signal at the initial stage of loading is generated. As the stress level increases, a large number of micro-fractures accumulate around the solid, and the number of cracks increases exponentially with the increase of the stress, and the micro-stress increment produces more defects. It is believed that each locked body is in the yield critical state, and any slight increase in stress will cause the interlocking development of the

fracture. When the micro-fracture exceeds the locked bodies stability threshold, the locked bodies yields and is constrained to the next locked bodies. When the last locked bodies of the rock mass breaks, due to the fracture is no longer restrained, the stress is instantaneously released, the crack rapidly expands, and the sample shows macroscopic cracking and transient instability, the stress at this time is the peak stress of the rock mass. Before the fracture of the last locked body, the critical fracture of each locked body corresponds to the development process of macroscopic cracks (will be shown as dynamic phenomena such as roof fracture, roof caving and slab before impact the engineering site), while the fracture of the last locked body indicates the occurrence of the main fracture (rock burst) of the sample.

### 4.2. Precursor Characteristics of Unstable AE of Granite Samples

The AE parameters (counting and cumulative energy) of granite samples showed obvious abnormal growth from sub-stability to instability stage (see Figure 4), especially the precursory phenomenon that the AE count shows intermittent spurt is the main feature. Whether it is laboratory rock mass fracture or engineering AE monitoring dynamic disaster, the AE response characteristics of this stage have significance reference for the early warning of rock mass dynamic disaster [19].

Before the macroscopic cracking of the sample, the AE mechanism is mainly based on the micro-yield such as the closure of the original crack of the sample and the micro-slip of the sample. As the grading loading progresses, the stress at the crack tip increases and the accumulated elastic energy increases after the original crack closing. When the energy of the crack tip (power P) is sufficient to overcome the energy (resistance R) that makes the locked bodies yield, the crack rapidly expands and the sample enters the sub-stability phase. At this stage, as the yield of the internal locked bodies of the sample increases, the strain continues to increase, and the mechanism of AE produces a transition to the macro crack. At this time, the AE signal not only depends on stress loading level, but also has a positive correlation with crack growth rate. In the crack propagation process (P > R), under the action of the locked bodies, resistance R regularly reduces the growth rate of crack propagation (a), and when the acceleration aˆ is a negative value, the power P and the resistance R are getting closer and closer, and when the power P and the resistance R are equal in value, the crack is stopped. At this stage, the AE signal enters a quiet period. As the stress level increases, the fracture power P will again reach the critical threshold yield of the solid lock (If the power P does not reach the critical value at this time, the crack derivative will stop the AE signal and enters the intermittent period). The locked bodies again yield plastically and the crack expands again.

When the axial load increases to the peak stress, the last locking solid that keeps the rock mass stable is about to fracture, and, at this time, the crack growth enters the instable growth. When the fracture density formed by a large number of internal cracks of the sample reaches the last solid locked fracture threshold, the sample will undergo a macroscopic fracture, instantly release a large amount of elastic strain energy, and the AE count reaches the maximum value [20].

From the theory of micro-fracture nucleation and multi-locked bodies fracture, it can be known that the macro-crack development process of granite is a micro-fracture clustering and forming a local fracture zone, and the local fracture zone is gradually expanded and nucleated to form a larger fracture zone. When the fracture density reaches the main fracture threshold of the sample, the sample is unstable. During this process, small cracks gradually aggregate and expand into large cracks. As the crack dimension is increased, the AE signal (counting and energy) has a sensitive high value response, which is reflected by the rising trend of the low value of the quiet phase of the AE signal, and the peak intensity is increased.

### 4.3. Discussion on the Characteristic of Different Fracture Modes of Granite Samples

As an effective means of damage assessment, AE is often combined with resistivity, electromagnetic radiation, ultrasonic and other non-destructive testing methods to evaluate the damage of structures and materials. There are two main ways to judge the crack type: firstly, the parameter method,

i.e., judging the internal fracture mode of the sample in accordance with RA value and AF value (RA = rising time/amplitude, AF = ringing count/duration). Studies indicate that AE signals with low AF and high RA values usually represent the generation or development of shear cracks, while high AF and low RA values represent the generation or development of tensile cracks. Secondly, quantitative inversion of moment tensor, i.e., achieving the properties, volume, spatial orientation and other parameters of the micro-crack after solving moment tenser by the first wave amplitude of six different spatial position sensors. However, this method is relatively complex due to involvement of the calibration test or deconvolution process of the sensor before the experiment as well as containment of the solution of the dynamic green functions. In this paper, the AE parameter method is used to study the relationship between the spatial distribution pattern of micro-cracks and its properties. Taking the pore rock as an example, the AE signal observation value during the rupture of the sample is shown in Figure 5, and the crack types over different spatial distributions according to the AE localization results are shown in Figure 6.

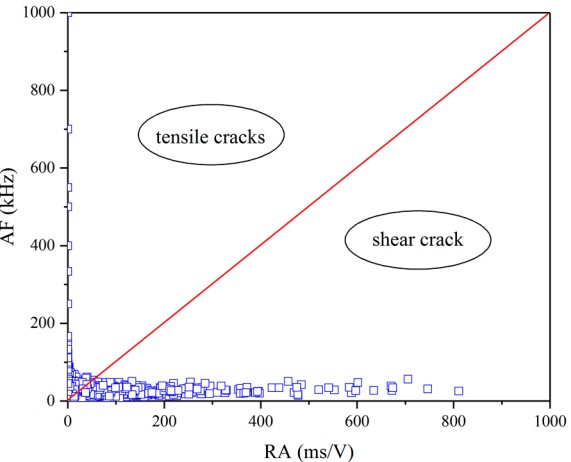

**Figure 5.** Distribution of AE parameters of RA versus AF.

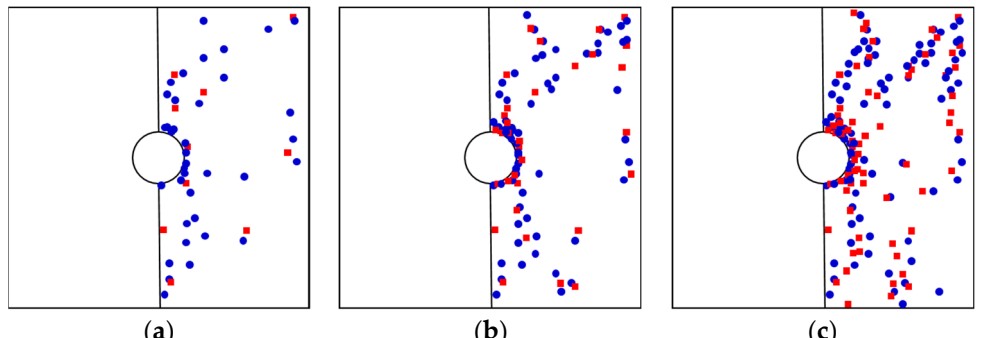

|            (a)            |            (b)            |            (c)            |

**Figure 6.** Spatial distribution of micro -cracks with an increase of stress. (**a**) 20% of tress; (**b**) 60% of stress; (**c**) 100% of stress.

It can be seen from Figure 5 that the fracture modes during the damage of the sample are tensile and shear composite crack and are mainly composed of tensile cracks. Take the right half of the hole rock as an example for analysis. As the stress increases, the crack spatial distribution is as shown in Figure 6. The tensile crack is indicated by a blue circle, and the shear crack is indicated by a red square.

In the early stage of the loading failure process (the stress is less than 20% of the peak stress), the micro-cracks of the hole rocks are distributed in a dispersed manner, and the damage of the rock sample was weak. AE events were mainly caused by the closure of the original defects in the sample. With the increase of stress (about 60% of the peak stress), micro-cracks begin to appear around the prefabricated holes, indicating that stress concentration began to appear around the holes, and shear

cracks began to appear above and around the holes, but tensile cracks still dominated at this time. With the increase of stress, the number of new shear cracks begin to increase and connect with each other and then gradually form macroscopic cracks. When the peak stress was reached, the connection between the old and new cracks led to the structural instability of the sample, which finally leads to macroscopic fractures appearing above and below the hole and penetrating the wall surface of the hole.

Certainly, the change of resistivity value can be used as a monitoring method to evaluate the damage status of samples under load. The continuum damage mechanics model, fiber-bundle model, the time-dependent and stress-dependent model can also be established to quantify the damage and these theoretical results are consistent with the experimental results. Gianni Niccolini et al. analysed the correlation between different modes of crack development and resistivity in the process of specimen damage by establishing a continuous damage mechanics model, and concluded that the relationship between damage and time is a power function, which provided theoretical support for the early warning of sample instability [21]. Bing Chen et al. jointly tested the resistivity and AE of concrete specimens, and verified the reliability of resistivity reflection on the specimen through AE technology, which indicates that the resistivity has a good correlation with the internal fracture mode of the specimen. This means that the damage and rupture process of the internal microstructure of the sample can be reflected by the change of resistivity, and the combined acoustic-electrical monitoring can more accurately characterize the damage of the sample [22].

Moreover, besides the resistivity being able to characterize the damage of the sample, other monitoring methods such as electromagnetic radiation and ultrasonic waves can also characterize the damage. Combining AE and electromagnetic radiation, the curve of the amplitude of electromagnetic radiation with time shows the internal damage of the sample, and the AE positioning can timely reflect the spatial distribution of the crack as well as reveal the extended path of the internal crack of the sample; therefore, the joint is deeply helpful to understand the fracture process and mechanical mechanism of the sample damage [23,24]. Combined with acoustic emission and ultrasonic testing, the damage state can be quantitatively evaluated through ultrasonic testing to obtain the damage distribution cloud chart under different stress levels. Wherein AE positioning technology can be used as an auxiliary means to monitor the real-time development status of cracks in the sample, quantitatively characterize the number and spatial distribution of cracks, and then obtain the dynamic migration path of damage and give the explanation of its mechanism. Therefore, combining with ultrasonic and acoustic emission monitoring methods can more fully understand the damage state of the sample and its instability evolution mechanism [25].

Therefore, integrated AE and some other monitoring methods, such as resistivity, ultrasonic, electromagnetic radiation, etc., can better identify the damage degree of the sample, recognize the damage mechanism of the sample, and enable more accurate disaster warning.

## 5. Conclusions

In this paper, the sequential characteristic law of AE in different graded loading was studied and the joint response characteristics of AE parameters and sample deformation instability were analyzed as well. The main conclusions are as follows:

(1) The sequential characteristics of the graded loading AE changing with stress are: the AE count in the compact stage is low-density trend and shows the loading enhancement feature; in the steady crack propagation stage, AE shows a steady rising trend with a small increase; the AE signal in the severe damage phase rapidly increases to a maximum.

(2) Rock-loaded fracture is a process in which small internal fractures connect to form large fractures and then prong to form the main fractures, that is, a large energy fracture occurs in a dense area of small energy events located by AE, and a gathering area of large energy AE events is the macro main fractures zone of the future. AE event localization can well characterize the evolution process of the sample fracture as well as explain the damage evolution process of the sample,

and the concentrated area of the AE large energy event coincides with the macro large crack position of the sample.

(3) There are similarities of AE signals during the loading process of intact rock samples and holey rock samples. However, due to the existence of hole, the AE counts and AE energy of the holey rocks are lower than those of intact rocks. Meanwhile, the stress level of holey rock is higher than that of the intact rock in the same stage at the end of the compaction phase due to the long compaction phase. In addition, the yield strength of the holey rock is less than that of the intact rock at the end of loading.

**Author Contributions:** Methodology and writing—review and editing, X.L.; Writing—original draft preparation, H.Z.; Data curation, C.Z., X.W., H.X., S.Y., and W.L.

**Funding:** This research was funded by the National Natural Science Foundation of China, Grant Nos. 51774280, 51634001.

**Acknowledgments:** The authors are grateful to all the people who help us in this paper.

**Conflicts of Interest:** The authors declare no conflict of interest.

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
