# Peer review of "Acoustic Emission Characteristics of Graded Loading Intact and Holey Rock Samples during the Damage and Failure Process"

_applsci, doi:10.3390/app9081595_

Round 1

Reviewer 1 Report

Throughout the article, there are missing spaces between words in different places, please repair.

Line 102: convert units to a metric system

Line 103: repair the unit

Line 122 and 123: Please indicate the total gain and threshold in Volts.

The names of the authors are written wrong, please correct (e.g. 9, 14), keep European style in references.

What is the difference between AE energy and AE count energy? These graphs are very similar.

In Figure 4b shows no holey in localisation graphs, how do you explain it?

Reviewer 2 Report

In this paper the Authors describe the relationship between rock damage and loads analyzed by means of AE monitoring.

The main results are: (1) Under the condition of grading loading, AE parameter increases with the increase of axial stress;  (2) The AE counts and energy are highly correlated with the fracture of the sample; (3) Compared with the intact rock sample, due to the pressure relief effect of the hole, the rock sample containing the hole takes a long time in the compaction stage and with higher load stress level.

The paper could be suitable for publication after some revisions. In the following a list of suggestions and requests of clarification are given.

As a first hint the manuscript is written in a quite good English. Only moderate changes are required.

The Referee suggests describing in more detail the experimental approach to make the manuscript more comprehensible to the Readers.

For example, at lines 115-116 the Authors declare that the sensors are fixed at the planned position on the surface of the sample through a special coupling agent. Which type of agent?  As a matter of fact, in certain condition, the use of silicon grease rather than a bi-component glue strongly attenuates wave propagation and so considerably reduces the number of detectable AE events.

More information about the coupling of the acoustic sensor to the specimen could be very useful.

Can the Authors describe which type of sensors have been used? Resonant or broadband? Typically, resonant sensors are more sensitive than broadband ones, which are characterized by flat response.

Moreover, at lines 122-123 the Authors assess that during the experiment, the preamplifier amplification was set to 40 dB, the threshold was 40 dB, and the sampling rate was 1 MSPS.

From the point of view of background noise, it is known that some of the problems in the use of acoustic sensors  is due to rapid changes of capacitance between conductors. In addition, flexing, twisting or transient impacts on coaxial cables could cause electromagnetic noises in the signal (with a spectrum from few Hz to tens THz). This phenomenon is generally called triboelectric effect and can induce false signals generation.

Many Authors tend to ignore this inconvenience, or they declare that no triboelectric effect is observed during their experimentation,  without however providing detailed information on how the latter is eliminated. In which way the Authors have overcome the problem of the triboelectric effect? It would be very helpful to improve the quality of the manuscript that the Authors spent a few words on this subject.

In addition, are the Authors able to provide some information about the identification of the cracking mode? It is well known that the connection between the fracture mode and the recorded acoustic waves depends on different factors like geometric conditions, relative orientation and propagation distance. In particular, the identification of the cracking mode may be done with the AE waves' rise time, the monitoring of peak amplitude, and the Average Frequency (AF) variation estimation (RILEM recommendations).

In the manuscript the Authors highlight the great potential of AE technique for studying the material internal damage degree, fracture evolution and macroscopic cracks generation and expansion

Moreover, in literature, several researches point out how the assessment of damage in structures and materials can be carried out also by both electrical resistance measurement and acoustic emission analysis.  In particular, the evolution of damage based on changing resistance shows agreement with theoretical predictions of continuum damage mechanics.

See and cites for example the following manuscripts:

 - Lemaitre J., Dufailly J. (1987) Damage measurements Engineering Fracture Mechanics Volume 28, Issues 5–6, 1987, Pages 643-661

- Chen B, Liu J (2008) "Damage in carbon fiber-reinforced concrete, monitored by both electrical resistance measurement and acoustic emission analysis". Constr Build Mater 22:2196.

- Niccolini G., Borla O., Accornero F., Lacidogna G., Carpinteri A. (2015) " Scaling in damage by electrical resistance measurements: an application to the terracotta statues of the Sacred Mountain of Varallo Renaissance Complex (Italy)". Rend. Fis. Acc. Lincei DOI 10.1007/s12210-014-0353-6.

An interesting point of argument would be if the Authors make some discussion regarding the above topic indicating how, in their opinion, the measurement carried out via acoustic emission sensors could be integrated with the electrical resistance monitoring and which type of additional information (in terms of damage assessment) they would expect to obtain.

The final comment of the Referee is that the manuscript is suitable for publication only after the above suggestions and revisions have been implemented.

Author Response

Response to Reviewer 2 Comments

Point 1: The Referee suggests describing in more detail the experimental approach to make the manuscript more comprehensible to the Readers.

For example, at lines 115-116 the Authors declare that the sensors are fixed at the planned position on the surface of the sample through a special coupling agent. Which type of agent?  As a matter of fact, in certain condition, the use of silicon grease rather than a bi-component glue strongly attenuates wave propagation and so considerably reduces the number of detectable AE events.

More information about the coupling of the acoustic sensor to the specimen could be very useful.

Response 1: We are so sorry for our unclear explanation, we have expended this part by:

According to the experimental scheme, the six resonant AE sensors (model R15α, peak frequency 150 kHz) should be fixed at the planned position on the surface of the sample through a bi-component glue before the experiment. The AE probes evenly were arranged along the diagonal of the two corresponding faces on the surface of the sample (holey sample should avoid AE probes arrangement on the surface with hole), which as shown in Figure 2.

In order to ensure the positioning accuracy, the lead-breaking experiment was used to test the accuracy and coupling of the positioning system before the experiment. The definition value and event locking value of AE events were adjusted repeatedly until the error distance between the location of the test point and the registration point was less than 2mm as well as the response amplitude of each sensor was more than 90dB. During the experiment, the preamplifier amplification gain was set as 40dB, the threshold value as 40 dB(100 uv), and the sampling rate as 1MSPS.

Figure 2. Schematic diagram of the experimental system

Point 2: Can the Authors describe which type of sensors have been used? Resonant or broadband? Typically, resonant sensors are more sensitive than broadband ones, which are characterized by flat response.

Moreover, at lines 122-123 the Authors assess that during the experiment, the preamplifier amplification was set to 40 dB, the threshold was 40 dB, and the sampling rate was 1 MSPS.

From the point of view of background noise, it is known that some of the problems in the use of acoustic sensors is due to rapid changes of capacitance between conductors. In addition, flexing, twisting or transient impacts on coaxial cables could cause electromagnetic noises in the signal (with a spectrum from few Hz to tens THz). This phenomenon is generally called triboelectric effect and can induce false signals generation.

Many Authors tend to ignore this inconvenience, or they declare that no triboelectric effect is observed during their experimentation, without however providing detailed information on how the latter is eliminated. In which way the Authors have overcome the problem of the triboelectric effect? It would be very helpful to improve the quality of the manuscript that the Authors spent a few words on this subject.

Response 2: thanks for this excellent comment, we have fixed it and explanation this part by:

Actually, the sensor we have been used is R15α resonant AE sensor. The R15α is a narrow band resonant sensor with a high sensitivity. The sensor cavity is machined from a solid stainless steel rod, making the sensor extremely rugged and reliable. The ceramic face along with a 30 degree chamfer to cavity electrically isolates the sensor cavity from the structure under test assuring a low noise operation.

This experiment was carried out in the electromagnetic shielding room which can effectively isolate external electromagnetic interference and environmental noise, and the shielding effect can be up to 85dB. In the experiment process, the movement of personnel was strictly controlled to ensure the experiment can be carried out in the condition of no interference to the greatest extent. Considering the electromagnetic noise interference, the sensor connectors were fixed on the test bench to avoid the flexing or twisting with coaxial-cable in the process of the experiment and the grounding wires were connected with experimental instrument to reduce unnecessary interference caused by electromagnetic noise during the acquisition of AE signals.

Fig1 R15α resonant AE sensor

Fig 2 The electromagnetic shielding room

Point 3: In addition, are the Authors able to provide some information about the identification of the cracking mode? It is well known that the connection between the fracture mode and the recorded acoustic waves depends on different factors like geometric conditions, relative orientation and propagation distance. In particular, the identification of the cracking mode may be done with the AE waves' rise time, the monitoring of peak amplitude, and the Average Frequency (AF) variation estimation (RILEM recommendations).

In the manuscript the Authors highlight the great potential of AE technique for studying the material internal damage degree, fracture evolution and macroscopic cracks generation and expansion

Moreover, in literature, several researches point out how the assessment of damage in structures and materials can be carried out also by both electrical resistance measurement and acoustic emission analysis.  In particular, the evolution of damage based on changing resistance shows agreement with theoretical predictions of continuum damage mechanics.

An interesting point of argument would be if the Authors make some discussion regarding the above topic indicating how, in their opinion, the measurement carried out via acoustic emission sensors could be integrated with the electrical resistance monitoring and which type of additional information (in terms of damage assessment) they would expect to obtain.

The final comment of the Referee is that the manuscript is suitable for publication only after the above suggestions and revisions have been implemented.

Response 3: thanks for the comments, we have fixed  it and explanation this part by

(1) There are two main ways to judge the crack type. Firstly, parameter method, i.e., judging the internal fracture mode of the sample in accordance with RA value and AF value (shown as Fig. 5). It can be seen from Fig. 5 that the fracture modes during the damage of the sample are tensile and shear composite crack and are mainly composed of tensile cracks. Take the right half of the hole rock as an example for analysis. As the stress increases, the crack spatial distribution is as shown in Fig. 6. The tensile crack is indicated by a blue circle, and the shear crack is indicated by a red square. Secondly, quantitative inversion of moment tensor, i.e., achieving the properties, volume, spatial orientation and other parameters of the micro-crack after solving moment tenser by the first wave amplitude of six different spatial position sensors. But this method is relatively complexity due to involve the calibration test or deconvolution process of the sensor before the experiment as well as contain the solution of the dynamic green functions.

 (2) In this paper the AE parameter method is used to study the relationship between the spatial distribution pattern of micro-cracks and its properties. The crack types over different spatial distributions according to the AE localization results are shown in Fig. 6. In the early stage of the loading failure process (the stress is less than 20% of the peak stress), the micro-cracks of the hole rocks are distributed in a dispersed manner, and the damage of the rock sample was weak. AE events were mainly caused by the closure of the original defects in the sample. With the increase of stress (about 60% of the peak stress), micro-cracks begin to appear around the prefabricated holes, indicating that stress concentration began to appear around the holes, and shear cracks began to appear above and around the holes, but tensile cracks still dominated at this time. With the increase of stress, the number of new shear cracks begin to increase and connect with each other then gradually form macroscopic cracks. When the peak stress was reached, the connection between the old and new cracks led to the structural instability of the sample which finally lead two macroscopic fractures appeared above and below the hole and penetrate the wall surface of the hole.

 (3) Certainly, the change of resistivity value can be used as a monitoring method to evaluate the damage status of samples under load. The continuum damage mechanics model, fiber-bundle model, the time-dependent and stress-dependent model can also be established to quantify the damage and these theoretical results are consistent with the experimental results. Gianni Niccolini et al. analysed the correlation between different modes of crack development and resistivity in the process of specimen damage by establishing a continuous damage mechanics model, and concluded that the relationship between damage and time is a power function, which provided theoretical support for the early warning of sample instability [21]. Bing Chen et al. jointly tested the resistivity and AE of concrete specimens, and verified the reliability of resistivity reflection on the specimen through AE technology, which indicates that the resistivity has a good correlation with the internal fracture mode of the specimen. That means the damage and rupture process of the internal microstructure of the sample can be reflected by the change of resistivity, and the combined acoustic-electrical monitoring can more accurately characterize the damage of the sample [22].

 (4) Moreover, besides the resistivity can characterize the damage of the sample, other monitoring methods such as electromagnetic radiation, ultrasonic waves can also characterize the damage. Combined AE and electromagnetic radiation, the curve of the amplitude of electromagnetic radiation with time shows the internal damage of the sample, and the AE positioning can timely reflect the spatial distribution of the crack as well as reveal the extended path of the internal crack of the sample, therefore, the joint is deeply helpful to understand the fracture process and mechanical mechanism of the sample damage [23-24]. Combined with acoustic emission and ultrasonic testing, the damage state can be quantitatively evaluated through ultrasonic testing to obtain the damage distribution cloud chart under different stress levels. Wherein, AE positioning technology can be used as an auxiliary means to monitor the real-time development status of cracks in the sample, quantitatively characterize the number and spatial distribution of cracks, and then obtain the dynamic migration path of damage and give the explanation of its mechanism. Therefore, combined with ultrasonic and acoustic emission monitoring methods can more fully understand the damage state of the sample and its instability evolution mechanism [25]

Therefore, integrated AE and some other monitoring methods, such as resistivity, ultrasonic, electromagnetic radiation, etc., can better identify the damage degree of the sample, recognize the damage mechanism of the sample, and enable more accurate disaster warning.

Fig.5 Distribution of AE parameters of RA versus AF

a)20% of stress                 b)60% of stress                     c)100% of stress

Fig.6 Spatial distribution of micro -cracks with an increase of stress

[21] Niccolini, G.; Borla, O.; Accornero, F.; et al. Scaling in damage by electrical resistance measurements: an application to the terracotta statues of the Sacred Mountain of Varallo Renaissance Complex (Italy)[J]. Rendiconti Lincei, 2015, 26(2):203-209.

[22] Chen, B.; Liu, J. Damage in carbon fiber-reinforced concrete, monitored by both electrical resistance measurement and acoustic emission analysis[J]. Construction and Building Materials, 2008, 22(11):2196-2201.

[23] Lemaitre, J.; Dufailly, J. Damage measurements[J]. Engineering Fracture Mechanics, 1987, 28(5):643-661.

[24] Wang, E.Y.; He, X.Q.; Wei, J.P. Electromagnetic emission graded warning model and its applications against coal rock dynamic collapses [J]. International Journal of Rock Mechanics and Mining Science, 2011, 48:556–564.

[25] Wang, X.R.; Liu, X.F.; W, E.Y.; et al. The Time-Space Joint Response Characteristics of AE-UT under Step Loading and Its Application[J]. Shock and Vibration, 2018, 2018:1-11.

Round 2

Reviewer 1 Report

Thank you for repair your text. I think that text is better at now. Good luck in the next experiments.

Author Response

Response to Reviewer 1 Comments

Point 1: Thank you for repair your text. I think that text is better at now. Good luck in the next experiments.

Response 1: Thank you very much for your excellent comment for my manuscript. Thank you and best regards.

Reviewer 2 Report

The Referee recognizes the effort done by the Authors to fully respond to the requests.

In general, the manuscript appears to be greatly improved if compared to the first version, and the requests made by the Reviewer seem to be well developed and integrated into the text.

The Referee only reports some typos within the text.

At line 124 please check the correctness of the verbal form "... should be fixed ...".

At line 133 probably the Authors intend mV instead of mn. Is it correct? Or is it a misunderstanding?

In figure 5 please amend the y-axis label: kHz instead of KHz.   

The final comment of the Referee is that the manuscript is suitable for publication after the above suggestions have been implemented.

Author Response

Response to Reviewer 2 Comments

Point 1: At line 124 please check the correctness of the verbal form "... should be fixed ...".

Response 1: We are so thankful for your sincere reminding. We have corrected it, please check.

Point 2: At line 133 probably the Authors intend mV instead of mn. Is it correct? Or is it a misunderstanding?

Response 2: Thank you very much for your excellent comment. We haven’t find “mn” at line 133.

Point 3: In figure 5 please amend the y-axis label: kHz instead of KHz.

Response 3: Thanks for your sincere reminding, we have corrected it.